# Comment on Bernard et al. Association between Urinary Metabolites and the Exposure of Intensive Care Newborns to Plasticizers of Medical Devices Used for Their Care Management. *Metabolites* 2021, *11*, 252

**DOI:** 10.3390/metabo11090596

**Published:** 2021-09-03

**Authors:** Rainer Otter, Angelika Langsch, Patrick Harmon, Scott C. Boito, Jan Mervart, Michael Grass, Nigel J. Sarginson

**Affiliations:** 1BASF SE, 67056 Ludwigshafen, Germany; angelika.langsch@basf.com; 2BASF Corp, Houston, TX 77079, USA; patrick.harmon@basf.com; 3Eastman, Kingsport, TN 37662, USA; sboito@eastman.com; 4DEZA, a.s., 757 01 Valašské Meziříčí, Czech Republic; J.Mervart@deza.cz; 5Evonik Operations GmbH, 45772 Marl, Germany; michael1.grass@evonik.com; 6ExxonMobil Petroleum and Chemical BV, 1831 Machelen, Belgium; nigel.j.sarginson@exxonmobil.com

## 1. Introduction

The recent publication “Association between Urinary Metabolites and the Exposure of Intensive Care Newborns to Plasticizers of Medical Devices Used for Their Care Management” by L. Bernard et al. (2021) [1] caught our interest.

The issue studied in this project is of utmost importance as most of the limited available data on the exposure of critically ill neonates were collected decades ago and represented the treatments used at that time.

Therefore, a new study reporting the results with 104 neonates has the potential to provide valuable new data.

However, some statements, hypotheses and recommendations in the publication by Bernard et al. merit detailed comments and require clarification.

## 2. Comments

Unfortunately, the publication and the supplementary materials are missing data and context necessary to draw any firm conclusions.

First, there is no information regarding the plasticiser content (*w*/*w*) of the medical devices (MD) studied in this project. The only information available is that Di(2-ethylhexyl) terephthalate (DEHT) was identified only in trace amounts in the medical devices and Diisononylcyclohexane-1,2-dicarboxylate (DINCH) was not identified in any of the devices; however, trace amounts of urinary metabolites of DINCH were identified with no further explanation.

As the authors already noted, using Fick’s diffusion law to predict plasticiser release into the gaseous phase may not be appropriate and may therefore lead to an overestimation of exposure by the inhalation route via respirators. The authors’ statement is further validated by the data in Table 1 showing no correlation between total plasticiser metabolites and the estimate for total site exposure. Given the neonates would only be exposed to plasticisers at the hospital, these data indicate there are deep flaws in the assumptions or methodologies of this publication. Available analytical data using microchambers and GC/MS for quantification of the saturated vapor concentrations under equilibrium conditions (Schossler et al., 2011) [2] show that gas phase concentrations >0.5 μg/m^3^ are not plausible for DINP or DINCH. Demir et al. (2013) [3] concluded that films plasticised with DEHT or DINCH exhibited the lowest mass loss among the phthalate and non-phthalate plasticizers used in soft-PVC films as confirmed by FTIR investigation.

Second, the publication does not provide any raw data or summary information on the physiological parameters of the neonates and does not indicate the type or frequency of procedures performed. Given the large variability in metabolite concentrations reported, this information would have been useful to develop more robust conclusions.

It is also important to note that plasticisers are an integral part of blood bag systems where their function is also to stabilize the red blood cells to allow storage of the blood units for up to 42 days (AuBuchon et al., 1988 [4]; Lagerberg et al., 2015 [5]). This is an essential use, so the wording *“contaminate the patient”* seems inappropriate and lacking an appropriate context, as without the plasticiser, not enough blood would be available for necessary transfusions.

### 2.1. Di(2-ethylhexyl) Phthalate (DEHP)

DEHP, which has been used for over 60 years in medical devices such as tubing and blood bags, has a harmonized classification (Repro Cat 1B) and labelling (H360FD) according to the CLP (Reg (EU) 1272/2008, Annex I) based on reproductive toxicity in rodents. The claim of the authors that DEHP would induce adverse effects in humans is unsupported—if that were the case, DEHP would be classified as Repro Cat 1A. Contrary to the authors, the EU Scientific Committee on Emerging and Newly Identified Health Risks (SCENIHR) [6] concluded: *“A review of the recent **epidemiological studies investigating DEHP exposure** associated with effects on testosterone production, breast tumour, hypospadias and cryptorchism, decreased anogenital distance, childhood growth and pubertal development, endometriosis, effect of DEHP metabolites on neurobehaviour, obesity, insulin resistance and type 2 diabetes, **were either inconclusive or inconsistent**”.*

According to the authors, the clinical trial part of the ARMED project was conducted between April 2014 and May 2015, and just before the French law [7] L-No 2012-1442 from 24 December 2012 came into force prohibiting the use of tubing containing DEHP in pediatrics, neonatals, and maternity as of 1 July 2015. Contrary to what the authors state, the new medical device regulation (MDR, Regulation (EU) 2017/745) [8] in Section 10.4. does not necessarily restrict the use of CMR substances since a justification is required pursuant to Section 10.4.2. Additionally, Section 10.4.3. refers to the mandatory guidelines on phthalates published by the EU Scientific Committee on Health, Environmental and Emerging Risks (SCHEER) in 2019 [9] and is effective starting from May 2021.

Figure 4 shows an increase in DEHP exposure by extracorporeal membrane oxygenation (ECMO) via an increase in urinary DEHP metabolites, which is in line with data published by Schneider et al. (1989) [10] and Karle et al. (1997) [11]. It is not clear why the authors state *“ECMO patients are highly exposed to DEHP, for which a direct association with urinary metabolites may be established”*. Increased DEHP exposure by ECMO is well known for 3 decades.

Finally, as noted above, DEHP is important as a plasticiser for life-saving medical devices and as such, a restriction of a substance in toys and childcare articles is of limited relevance for the use of this substance.

### 2.2. Diisononyl Phthalate (DINP)

The authors claimed data gaps for DEHP substitutes (DINCH, DEHT) and postulated endocrine-disrupting properties for DINP, while ignoring the recent risk assessments and the decision of the ECHA risk assessment committee (RAC-44)) from 2018 [12] that concluded no classification for DINP for either effects on sexual function and fertility or for developmental toxicity. The results of a recent state of the art study published by van den Driesche et al. (2020) [13] perfectly support this decision.

The authors report the use of the plasticiser DINP for transfusion units in direct blood contact. The European Pharmacopoeia lists DEHP and four further substitutes, namely DINCH, DEHT, TOTM, and BTHC, but not DINP. DINP is not intended to be used in medical applications with direct contact to blood due to the lack of data on the intravenous route following repeated exposure. A benefit–risk assessment for DINP and a justification for the use of DINP in such an application seems to be premature given that a study on the intravenous route, i.e., with direct blood contact, is not available for DINP. The use of the oral DNEL derived by the REACH registrants for DINP and published on the ECHA website for the evaluation of the internal exposure via transfusion seems not to be appropriate. On the oral route, the primary cleavage product of DINP, the monoester MINP is systemically available, while via transfusion the diester is systemically available (Campbell et al., 2020) [14]. A more detailed discussion to understand the DINP-related urinary cx-MINP concentrations normalized to creatinine (Supplementary File S2) and the association with the number of transfusions would have been helpful. Further, the authors would be well advised to consolidate and correct the chemical names of several metabolites in Figure 7 as well as in the Supplementary File 2 (i.e., cx-MINP, and below it reads for the same metabolite, cx-MeMINP, which is wrong, as the more correct nomenclature for this metabolite would mono—(4-methyl-7-carboxyheptyl) phthalate).

### 2.3. Diisononyl Cylcohexane-1,2-dicarboxylate (DINCH) and (Di(2-ethylhexyl) Terephthalate (DEHT)

The authors acknowledge that DINCH was not identified as a component in any medical device analysed in this study. Given the minute DINCH metabolite levels measured in urine could be background exposure, the authors make some rather incredible claims about the safety of DINCH.

The statement with reference to NICNAS [15] that DINCH exposure in Australia would be “close to the tolerable intake” is unsupported—the respective NICNAS report does not give any indication for this. In fact, Gomez Ramos et al. (2016) [16] report an average level of 3.9 ng/mL for the DINCH-specific urinary metabolite (monohydroxyisononyl cyclohexanoate) MHINCH based on the measurement of urinary metabolites in 24 pooled samples; each pool consisted of 100 individuals. Reverse dosimetry results in an exposure of the general public in Australia which is far below 1 µg/kg bw/day, which is by a factor of 10^4^, i.e., 10,000 times lower than the TDI set by EFSA of 1 mg/kg bw/day.

A potential risk from background exposure to DINCH can be assessed using the NOAEL of 300 mg/kg bw/day for the intravenous route (David et al., 2015) [17]. Reverse dosimetry for the values given in Figure 3, taking into account standard parameters for neonates, shows that the exposure of neonates regarding DINCH is more than 1 million-fold lower than the TDI oral (1 mg/kg bw/day) by EFSA, i.e., the minimal background exposure reported for the neonates is negligible and presents no risk.

The conclusions drawn by the author rely on in silico testing rather than using reliable data gathered from in vivo experiments. Sheikh et al. (2016) [18] report the binding of DEHT, DINCH, and TOTM with 25–30 amino acid residues of the sex hormone-binding globulin (SHBG) in an in silico docking study based on structural data of SHBG available from a Protein Data Bank (PDB; http://www.rcsb.org/). Such studies are most often of very limited value, especially when in vivo results published by Furr et al. (2014) [19] indicate no anti-androgenic effect for DINCH and DEHT as measured by fetal testosterone production. A lack of estrogenic and progestogenic activity was published by Wenzel et al. (2021) [20], and lack of estrogenic activity was also confirmed by U.S. EPA [21] for DINCH and DEHT.

While the authors cite from the U.S. CPSC evaluation report (2018) [22] on DEHT *“For DEHT, recent conclusions are that it is not clear whether this plasticizer causes reproductive toxicity or not”*, in fact both the U.S. CPSC evaluation report from 2018, as well as the updated CPSC report 2019 [23], conclude that DEHT is neither a reproductive toxin nor does it cause malformations. Indeed, there were some questions raised relating to variations and reduced fetal weights secondary to decreased maternal food consumption; however, such effects need to be evaluated according to the respective OECD evaluation guidelines [24]. Interestingly, while this publication was submitted to the journal on 22 March 2021, several co-authors of L. Bernard recognized already on 29 November 2020, when submitting the publication by Kambia et al., 2021 [25], the lack of reproductive toxicity of DEHT, stating: *“Di-(2-ethylhexyl) terephthalate (DEHT) is an interesting candidate due to its lower migration from PVC and its lack of reprotoxicity”.*

The authors state that DEHT exposure was very low, but DEHT was only found in trace levels in the medical devices (MD); i.e., it was not the plasticiser in the devices.

Unfortunately, the authors present selective data by referring to in vitro and in silico data published by Kambia et al. (2019) [26] reporting a potential estrogenic activity/endocrine disruptive activity of the 5OH-MEHT metabolite of DEHT. However, they did not mention that relevant in vivo studies (Gray et al., 2000 [27]; Faber et al., 2007 [28]) did not show evidence for androgenic or estrogenic activity in vivo. The conclusions/hypotheses presented by the authors are not reflected by the data presented and/or result from a one-sided selection of references that do not reflect the entirety of available data.

The rationale for the recommendation to use DEHT for “pharmaceutical firms” seems ambiguous—the project did not disclose any pharmaceutical products that contain DEHT. However, exposure to DEHT seems to be more than 10 times higher than that of DINCH. Further, DEHT was only identified as a minor compound in the medical devices studied. The available data as reviewed by the U.S. CPSC and others suggest that DEHT is an attractive alternative to DEHP; however, it is hard to understand how this conclusion in Bernard et al. was reached based on the data presented in the paper.

## 3. Summary

In order to come to firm conclusions on the suitability of certain plasticisers for medical devices, reliable hazard data of the plasticisers, preferably on the intended route of exposure, and exposure data for the respective applications are needed.

The conclusions from ARMED as presented in the publication by Bernard et al. (2021) are inconsistent with the published data for the substances and published official assessments/evaluations.

## Data Availability

Data available in a publicly accessible repository. The data presented in this comment are openly available, see list of references.

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
