# Peer review of "Comment on Bernard et al. Association between Urinary Metabolites and the Exposure of Intensive Care Newborns to Plasticizers of Medical Devices Used for Their Care Management. Metabolites 2021, 11, 252"

_metabolites, 2021, doi:10.3390/metabo11090596_

Round 1

Reviewer 1 Report

The comments by Otter et al are too technical for a clinical neonatologist to comment on the substance of the matter.

In any case, Bernard's manuscript produced some interesting results, and now Otter refutes them with bibliography and legislation.

I believe that the discussion is of great interest and it can, undoubtedly, provide reasons for experts to value both texts.

I repeat that, as a clinical neonatologist, I do not consider myself capable of judging who is right, but I do believe that the topic and the discussion are appropriate and interesting.

Regarding the forms of the text, I think Otter's comment is hard but correct, defending his own views and providing references. For this reason, I believe that these comments can be published and should be answered by Bernard et al.

Author Response

Dear reviewer 1, thank you for taking the time to complete your review. We appreciate your comments.

Reviewer 2 Report

I do not have concerns.

Author Response

Dear Reviewer,

thank you very much for taking the time to review our comment and for your support 

Reviewer 3 Report

  1. Regarding the comment by Otter et al., “….(DINCH) was not identified in any of the devices, however, trace amounts of urinary metabolites of DINCH were identified with no further explanation.”: Bernard et al., Page 3 of 16, paragraph 2.3: “On the contrary, unexpectedly, DINCH metabolites were found in all urine samples, suggesting another route of exposure than from MD.” ALSO in Discussion, Bernard et al., mention: “As we found trace levels of its metabolites in the urine, we suggest that the contamination is provided by another source, such as incubators (by inhalation exposure, another route of exposure to plasticizers [36–38]), babies’ diapers [39–41], or the mother-to child transmission during pregnancy (this latter being quite unlikely because of the fast elimination of DINCH metabolites in the organism [42,43], while we found levels in the urine of the babies each study day), or even during breastfeeding.“ Taking into consideration that Bernard et al., are giving explanations for their biomonitoring findings, and throughout the article it is not considered that the sole source of exposure to phthalates is medical devices, it is suggested that the certain comment by Otter et al., has already been answered.

  1. Regarding the comment by Otter et al., “Second the publication does not provide any raw data or summary information on the physiological parameters of the neonates and does not indicate the type or frequency of procedures performed. Given the large variability in metabolite concentrations reported, this information would have been useful to develop more robust conclusions.” The large variability in biomonitoring results is being commonly met in biomonitoring studies and it is depending on several parameters, such as the time of sampling or varying sources of exposure, and they cannot be always surpassed. Furthermore, Bernard et al., mention that this variability is “depending on the center” and it is absolutely correct as different environments, practices, devices and instrumentation could have different impact on patients’ body burden to the compounds.

  1. Regarding the comment by Otter et al., “The claim of the authors that DEHP would induce adverse effects in humans is unsupported…”, there are undoubtedly several studies in literature with findings regarding the potential health impact of phthalates including DEHP, in males and females. Indicatively, studies that associate exposure to phthalates and effects only in children are:

Arbuckle et al., 2016/NeuroToxicology, 54, 89-98; Bertelsen et al., 2013/Environ. Health Perspect. 121 (2), 251–256; Park et al., 2014/Int. J. Environ. Res. Public Health 11 (7), 6743–6756; Trasande and Attina 2015/Hypertension 66 (2), 301–308; Valvi et al., 2015/Environ. Health Perspect. 123 (10), 1022–1029; Wang et al., 2013/PLoS One 8 (2); Xie et al., 2015/Mol. Cell. Endocrinol. 407, 67–73; Zhang et al., 2015/Environ. Int. 83, 41–49

and the findings refer to various effects including associations with disorders and syndromes, blood pressure, body mass index, delayed growth and puberty in males, speed up breast development and earlier menarche onset in females, testosterone levels and respiratory problems.

Given the existence of studies that do not support such findings, authors are correctly being mentioned to potential health effects.

  1. Regarding the comment by Otter et al., “The authors claimed data gaps for DEHP substitutes (DINCH, DEHT) and postulated endocrine disrupting properties for DINP, while ignoring the recent risk assessments and the decision of the ECHA Risk assessment committee (RAC- 93 44)) from 2018 [12] that concluded no classification for DINP for either effects on sexual function and fertility, or for developmental toxicity.” Despite the decision of the ECHA that “there is no evidence for effects of DINP on development in humans,” according to the United States Consumer Product Safety Commission (https://www.federalregister.gov/documents/2017/10/27/2017-23267/prohibition-of-childrens-toys-and-child-care-articles-containing-specified-phthalates) : “prohibit children's toys and child care articles that contain concentrations of more than 0.1 percent of diisononyl phthalate (DINP), diisobutyl phthalate (DIBP), di-n-pentyl phthalate (DPENP), di-n-hexyl phthalate(DHEXP), and dicyclohexyl phthalate (DCHP)” AND “The CPSIA authorizes the Commission to “declare any children's product containing any phthalates to be a banned hazardous product” if such action is necessary to protect the health of children”.

In cases where data are still unclear such as for high molecular weight phthalates and phthalates substitutes 1) more research studies should be encouraged to be conducted and no attempts to restrict scientific research can be acceptable 2) Regulations should be continuously revised and 3) “no classification” of a compound (such as in the case of DINP) cannot be translated as “safe compound” especially when there are other organizations and agencies that highlight the need to avoid the certain compound for health safety.

  1. Regarding the comment by Otter et al., “Unfortunately, the authors present selective data by referring to in vitro and in silico data published by Kambia et al. (2019) [26] reporting a potential estrogenic activity/endocrine disruptive activity of the 5OH-MEHT metabolite of DEHT. However, they did not mention that relevant in vivo studies (Gray et al., 2000 [27]; Faber et al., 2007 [28]) did not show evidence for androgenic or estrogenic activity in vivo. The conclusions/hypotheses presented by the authors are not reflected by the data presented and/or result from a one-sided selection of references that do not reflect the entirety of available data.” Authors present more recent data about the potential health effects (2019) rather than earlier information (2000, 2007). Furthermore, the discussion about potential effects in text answers the relevant comment as well as the fact that these studies are not included in text does not alter the findings and the biomonitoring results of the present study.

Author Response

Dear reviewer 3, thank you for taking the time to review and provide your detailed comments, that we appreciate. 

We recognize that we have some differing opinons regarding validity of studies you cited - but that reflects the spirit of a healthy scientific dispute.

Please find our detailed responses to each of your comments in the attached file. 

Reviewer 4 Report

Otter et al. comment on the publication “Association between Urinary Metabolites and the Exposure of Intensive Care Newborns to Plasticizers of Medical Devices Used for Their Care Management” by L. Bernard et al. published in Metabolites in 2021. They describe and document valid concerns and fully disclose potential conflicts of interest. I believe the comment merits publication and an opportunity for clarification and rebuttal by Bernard et al. 

Minor typos: 

l. 47 ...indicates there is are deep flaws....

l. 102 ...be premature given that a study on as a study on the intravenous route....

Throughout the text: 'in silico' in italic or non-italic throughout the text

Author Response

Dear Reviewer 4, thank you for the time spent to review our comment submitted for publication, especially thank you very much for your kind support spotting the typos. All of them (together some others newly identified) are now corrected.

Round 2

Reviewer 3 Report

Ι have nothing to add